



# Temperature-dominated spatiotemporal variability in snow phenology factors on the Tibetan Plateau from 2002 to 2021

Jiahui Xu[1,2,3], Yao Tang[1,2,3], Linxin Dong[1,2,3], Shujie Wang[4], Bailang Yu[1,2,3], Jianping Wu[1,2,3], Zhaojun Zheng[5], Yan Huang[1,2,3,*]

[1]Key Laboratory of Geographic Information Science, Ministry of Education, East China Normal University, Shanghai, 200241, China
[2]School of Geographic Sciences, East China Normal University, Shanghai, 200241, China
[3]Key Laboratory of Spatial-temporal Big Data Analysis and Application of Natural Resources in Megacities, Ministry of Natural Resources, Shanghai, 200241, China
[4]Department of Geography, Earth and Environmental Systems Institute, Pennsylvania State University, University Park, PA, 16802, USA
[5]National Satellite Meteorological Center, Beijing, 100081, China

*Correspondence to*: Yan Huang (yhuang@geo.ecnu.edu.cn)

**Abstract.** A detailed understanding of snow cover and its possible feedback on climate change on the Tibetan Plateau (TP) is of great importance. However, spatiotemporal variability in snow phenology (SP) and its influencing factors on the TP remain unclear. Based on the daily gap-free snow cover product (HMRFS-TP) with 500 m resolution, this study investigated the spatiotemporal variation in snow cover days (SCD), snow onset date (SOD), and snow end date (SED) in the TP from 2002–2021. A Structural Equation Model was used to select the factors affecting SP as well as to quantify the direct and indirect effects of meteorological factors, geographical location, topography, vegetation greenness, and atmospheric pollution factors on SP. The results indicated that the spatial distribution of SP on the TP was extremely uneven and exhibited notable temporal heterogeneity. SP showed vertical zonality influenced by elevation (longer SCD, earlier SOD, and later SED at higher elevations). Meanwhile, their interannual variations tended to decrease, delay, and delay slightly from 2002 to 2021. In particular, the interannual variation in SP also had an elevation-dependent pattern below 5800 m. Meteorological factors had direct and indirect effects, vegetation greenness only had a direct impact, and geographical location, topography, and atmospheric pollution only indirectly affected SP. Undoubtedly, meteorological factors were the dominant factors in particular temperature. However, the influence of other factors cannot be ignored. As two important factors, the relative importance of temperature versus precipitation to SP shifted across elevation. This study contributes to the understanding of snow variation in response to global warming over the past two decades by providing a basis for predicting future environmental and climate changes and their impacts on the TP.



## 1 Introduction

Rapid accumulation and melting of snow make it one of the most active natural materials on the Earth's surface (Gutzler and Rosen, 1992; Ma et al., 2023). The Tibetan Plateau (TP) is one of the most sensitive regions to climate change because of its unstable alpine ecosystems and fragile natural environment (Huang et al., 2017). In the past 50 years, temperature has risen at a rate of about 0.37 °C per decade on the TP (You et al., 2021). Rising temperatures inevitably affect snow accumulation and melting processes, exemplified by variations in snow phenology (SP), including snow cover days (SCD), snow onset date (SOD), and snow end date (SED) (Chen et al., 2015; Guo et al., 2022). Variations in SP can in turn affect terrestrial ecosystems and feedback on regional climate (Cherkauer and Sinha, 2008). Early snowmelt also causes substantial variability in the onset date and amount of snowmelt runoff, increasing the incidence of disasters (e.g., floods) (Fyfe et al., 2017). A detailed understanding of the variability in SP and its possible feedback on climate change is of great significance to the hydrological cycle (Kraaijenbrink et al., 2021), ecological balance (Keyser et al., 2022), and societal security (Wang et al., 2017) in the TP.

There are typical interannual variations in the SP on the TP. Ground-measured measurements and remote sensing data have been extensively used in recent decades to better understand these changes (Ma et al., 2022; Notarnicola, 2020). Remote sensing is beneficial in high-elevation area where few ground-measured measurements are obtainable (Huang et al., 2022a). Passive microwave satellite (e.g., SMMR, SSM/I, and AMSE-R) and multispectral satellite (e.g., Landsat, Sentinel-2, and MODIS) data are widely used to retrieve snow information and reveal its variability (Chen et al., 2018). Although passive microwave data is available over an extended period, the spatial resolution (~10−25 km) is relatively coarse (Huang et al., 2017). Landsat and Sentinel-2 have a high spatial resolution (~10−30 m) but a relatively coarse temporal resolution (10 or 16 days), inadequate for monitoring temporal variability in SP (Huang et al., 2022a). Currently, the most widely available snow data used for the TP is the daily MODIS snow product (Hall et al., 1995), which has revealed that the SP on the TP has changed considerably in the last two decades, characterized by a significant decrease of SCD and delay of SOD, and a non-significant advance of SED (Ma et al., 2023; Tang et al., 2022; Wang et al., 2017). In addition, owing to the complex topography of the TP, elevation may have a particular impact on SP. However, it is still unclear whether interannual variations in SP are elevation-dependent (Ma et al., 2023; Wang et al., 2021). Moreover, since the original daily MODIS images have data gaps due to cloud cover and relatively low estimation accuracy due to complex topography of the TP (Huang et al., 2022b), most previous studies used relatively simple methods to fill data gaps and did not consider topographic effects. Using snow products without topographic corrections may result in inaccurate SP extraction, further affecting the analysis of interannual variations. Huang et al. (2022b) generated a daily gap-free snow product on the TP by employing a Hidden Markov Random Field (HMRF) framework to MODIS product (HMRFS-TP). The data gaps were filled by optimally integrating spatiotemporal, spectral, and environmental information. Accuracy of HMRFS-TP was significantly improved over the complex topography, providing spatiotemporally continuous information on snow distribution over an extended period with high accuracy.

Regarding the influencing factors on SP of the TP, previous studies have highlighted a complex and heterogeneous situation in which SP is driven by meteorological factors, topographic conditions, geographical location, and vegetation



greenness (Pulliainen et al., 2020; Qi et al., 2021; You et al., 2020). Temperature and precipitation are the two dominant
meteorological factors affecting SP (Li et al., 2022; Tarca et al., 2022). Temperature defines precipitation-to-snowfall ratio
and snowpack melt (Ishida et al., 2019). SP is also closely associated with wind and humidity; however, their quantitative
influence on SP on the TP has not been discussed further (Ishida et al., 2019; Ma et al., 2022; Xie et al., 2017). Topographic
characteristics (Guo et al., 2022) and vegetation greenness (Qi et al., 2021) are also considered responsible for SP. Elevation
is an essential factor that influences SP because snow accumulates more at higher elevations. Steeper slopes may receive less
snow than flatter ones due to gravitational pull (Li et al., 2011). Vegetation greenness changes the snow redistribution process
by intercepting snow, and affects spatial pattern and melt rate of surface snow by regulating the solar radiation budget balance
(Barrere et al., 2018). Recent studies (Kang et al., 2019) have found that the deposition of atmospheric pollution (e.g., black
carbon) can influence snowfall as well as snowmelt, and thus should be regarded as an essential physical factor affecting snow
variation on the TP. For instance, increased black carbon and dust reduced the SCD by 3–4 days on the TP (Zhang et al., 2018).

Although much research has examined the variability in the SP and its influencing factors on the TP, uncertainties and
limitations remain. Previous studies have mainly focused on the response of SP to climate change, but few studies have
quantitatively explored other associated factors influencing SP. Rather simple correlation analyses were primarily used to
explore relationships between SP and its influencing factors, without consideration of the intercorrelations between factors.
Associated factors may have mediating effects on SP, which cannot be explained by simple correlation analyses. For example,
temperature can directly affect SP, as well as indirectly affect SP by controlling the distribution of vegetation greenness (Qi et
al., 2021). The specific process by which each factor influences SP (i.e., directly, or indirectly) has not been elucidated. The
Structural Equation Model (SEM) is a suitable method for studying such coupling relationship, which can explain the direct
and indirect relationships between factors (Grace et al., 2010; Shen et al., 2022; Zhang et al., 2022). Here we adopt this
approach to explore the mediating effects between SP and associated factors.

In this study, we conducted a comprehensive investigation on the spatiotemporal variability in SP on the TP over the
past two decades (2002–2021), based on the daily gap-free HMRFS-TP dataset, and further quantified the influence of
associated factors on SP. The main goals are to (1) calculate SP parameters (SCD, SOD, and SED), and analyze their
spatiotemporal variability; (2) examine whether interannual variation in SP on the TP is elevation-dependent; and (3) reveal
the direct and indirect effects of associated factors on SP.

# 2 Study area and data

## 2.1 Study area

The TP (26°00′12″N-39°46′50″N, 73°18′52″E-104°46′59″E) has an average elevation of over 4000 m and an area of
approximately $2.5 \times 10^6 \, \text{km}^2$ (Figure 1). Under the background of global warming, enormous changes have taken place in the
TP cryosphere. Temperature in its permafrost region is rising at about three times the speed of global warming (Wang et al.,
2022). As a topographic barrier, the TP forces air mass up and produces cooling effects (Wu et al., 2015), further influences

the thermodynamic properties of atmospheric circulation, strength and duration of Asian monsoon systems, global climate, and energy budge (Fan et al., 2019). The southeast is warm and moist while the northwest is cold and dry in the TP (Wang et al., 2023). It is the source of several prominent Asian rivers (e.g., the Yellow River, Yangtze, Indus, and Mekong), which are primarily fed and regulated by meltwater from glaciers and snow (Chen et al., 2022). Snowmelt water is used for livelihoods

and irrigation in the region and its surrounding areas. The change in SP affects the vegetation greenness of the plateau and its surrounding areas.

## 2.2 Datasets

### 2.2.1 Snow cover product

The daily gap-free HMRFS-TP at 500 m resolution from 2002 to 2021 was used to extract SP (Huang et al., 2022b). The

HMRF technique produces a dataset (Huang et al., 2018) that optimally couples spatiotemporal, spectral, and environmental information to fill data gaps in original MODIS images. Accuracy of the HMRFS-TP is 91.36% and 98.29% based on snow maps derived from Landsat images and *in situ* observations, respectively. The estimation accuracy is notably improved in snow transitional periods and complex topography with higher elevations and sunny conditions. To determine the SP (including SCD, SOD, and SED), we specify the snow season from 1 September of the last year to 31 August of the current

year (Tang et al., 2022). The SP for 19 snow seasons from 1 September 2002 to 31August 2021 was extracted based on the long-term HMRFS-TP dataset.

### 2.2.2 Meteorological data

To explore the influence of meteorological factors on SP, monthly temperature and precipitation from 2002to 2021 with 1 km resolution were used (Peng et al., 2019). This dataset was generated from two global high-resolution climate data through a

delta spatial downscaling model. Monthly humidity, wind speed, and downward shortwave radiation with 0.1° resolution from 2002 to 2018 were created by coupling *in situ* observations, remote sensing, and reanalysis dataset (He et al., 2020). All monthly meteorological data were averaged for each snow season and resampled to 500 m to maintain consistency with the snow product.

### 2.2.3 Digital Elevation Model (DEM)

The 30 m DEM data, available from the USGS Earth Explorer (https://earthexplorer.usgs.gov), was resampled to 500 m resolution. Topographical parameters including elevation, slope, and aspect were calculated from it.

### 2.2.4 Land surface reflectance product

Compared with other vegetation indexes, the Normalized Difference Greenness Index (NDGI) (Yang et al., 2019) has been proven to more accurately represent the vegetation greenness growth status in snow-covered areas such as the TP (Xu et al.,



2022a; Xu et al., 2022b). Here, we calculated the NDGI based on MODIS Terra surface reflectance MOD09A1 with 500 m resolution and an 8-day repeat cycle from 2002 to 2021 (Vermote, 2021). The bands 4, 1, 2 of MOD09A1 were used to calculate the NDGI. The maximum complex method was implemented to filter noise and fill data gaps owing to cloud and atmosphere in NDGI series (Xu et al., 2022b). The annual average NDGI time-series composite data for each snow season were obtained.

**2.2.5 Atmospheric pollution data**

We used two essential atmospheric pollution datasets, Black Carbon (BC) emissions (g km$^{-2}$) and Aerosol Optical Depth (AOD) concentration ($\mu$g m$^{-3}$), to explore the interaction between atmospheric pollution and SP. The monthly global BC emission estimates covered 73 detailed sources at a 0.1° resolution from 2002 to 2017 (Xu et al., 2021). The monthly AOD dataset at a resolution of 1 km was obtained by synergistically integrating multimodal aerosol data (Bai et al., 2022). We

converted all monthly BC emissions and AOD concentration data to the annual average during the snow season and resampled them to 500 m resolution.

**2.2.6 Auxiliary data**

Snow depth recorded from meteorological stations was used as the reference value (ground-observed SP) to validate the accuracy of the satellite-derived SP parameters. Due to instrument machine failure or man-made errors, not all meteorological

stations (approximately 137) on the TP recorded snow depth values every day. This resulted in many data gaps during the snow season, which were not sufficiently continuous to extract ground-observed SP. Therefore, 24 meteorological stations with daily records were selected for validation (Figure 1).

**3 Methods**

  Figure 2 illustrates the workflow of this study. The SP parameters (SCD, SOD, and SED) were first extracted from the daily

HMRFS-TP dataset, and their spatiotemporal variations were analyzed. We then adopted SEM to identify the factors that were relatively important to SP: meteorological factors, topographic conditions, geographical location, vegetation greenness, and atmospheric pollution. The direct/indirect effects of these factors on SP were also analyzed using SEM. We further quantitatively discuss the effects of these factors on SP.

**3.1 Snow phenology extraction and accuracy assessment**

Three parameters, SCD, SOD, and SED, were derived to describe SP in each snow season and were calculated from the daily HMRFS-TP dataset at the pixel level. SCD is the sum of days when a pixel is covered by snow across the snow season. SOD is the first date when a pixel is covered with snow stay at least 5 days in a snow season. SED is the last date on which a pixel was identified as snow for 5 consecutive days (Tang et al., 2022). Using a threshold value of 5 days can reduce the influence





of frequent short-term variability (e.g., snowmelt and accumulation in early spring and late autumn), which has been a widely

used threshold on the extraction of SOD and SED (Guo et al., 2022; Wang et al., 2017; Xu et al., 2022b).

The accuracy of the extracted SP was validated using ground-observed SP from snow depth measured by meteorological stations. The ground-observed SP was calculated as follows. First, all recorded snow depth values were reclassified as snow or no snow based on a 3 cm threshold (Huang et al., 2022a; Huang et al., 2022b). For each snow season, meteorological stations that recorded more than 200 days of snow depth were selected (Hao et al., 2022; Zhao et al., 2022). Days without snow depth

records at the selected stations were supplemented with remote sensing images (Landsat series, Sentinel-2, etc.) of these days. Finally, stations with fewer than 20 snow-covered days and fewer than 5 consecutive snow-covered days during the snow season were excluded. Fifty-six ground-observed SP from 24 meteorological stations were collected for accuracy assessment.

### 3.2 Trend analysis

To explore the interannual variation trend and trend significance in SP from 2002 to 2021, the Theil-Sen non-parametric

regression (Sen, 1968) and Mann-Kendall (M-K) tests (Hirsch et al., 1982) were applied to each pixel. The Theil-Sen method has the advantage of dealing with non-normally distributed data, and is robust against outliers compared with traditional linear regression (Theil, 1992).

$$\beta = Median\left(\frac{x_j - x_i}{j - i}\right),\ 1 < i < j < n, \tag{1}$$

where $\beta$ is the trend slope; a positive value denotes the trend of SP as a delay/extension, and a negative value assumes SP is

advanced/delayed. $n = 19$, $x_i$ is the $i^{\text{th}}$ value in 19 years. Same as $x_j$ and $j^{\text{th}}$.

The M-K test is a non-parametric approach for monotonic trend that has been used for trend detection of hydrological and meteorological time series (Qi et al., 2021). The Z value assumes the temporal trend is statistically significant:

$$Z = \begin{cases} \frac{S-1}{\sqrt{Var(S)}}, & if\ S > 0 \\ 0, & if\ S = 0 \\ \frac{S+1}{\sqrt{Var(S)}}, & if\ S < 0 \end{cases}, Var(S) = \frac{n(n-1)(2n+5)}{18}, \tag{2}$$

$$S = \sum_{i=1}^{n-1}\sum_{j=i+1}^{n} \begin{cases} 1, & if\ x_j > x_i \\ 0, & if\ x_j = x_i, 1 < i < j < n, \\ -1, & if\ x_j < x_i \end{cases} \tag{3}$$

when $|Z| > 1.28$, 1.64, and 2.32, the tests were significant at levels of 0.1, 0.05, and 0.01, respectively.

Since snow cover may not be present each year, to evade unnatural results caused by a small amount of data, only the pixels with at least recorded 6 years for the SP were used for the interannual trend evaluation (Xu et al., 2022b).





### 3.3 Structural equation model

We used SEM to identify relatively important factors and quantify their direct and indirect effects on SP. It is a multivariate
collection of methods that can simulate the interaction between various factors at the same time, supplying a framework for
extrapolating cause-effect relationship and revealing direct/indirect relations between independent and dependent variables
(Grace et al., 2010). SEM includes Covariance Based-Structural Equation Model (CB-SEM) and the Partial Least Square
Structural Equation Model (PLS-SEM) (Venturini and Mehmetoglu, 2019). Contrasted with CB-SEM, PLS-SEM focuses on
mining sample information and can reflect the nature and structural characteristics of objects as much as possible, making it
more suitable for exploring newly constructed structural models (Hair et al., 2011; Ringle et al., 2012). More importantly,
PLS-SEM is a nonparametric model that does not require a normal distribution of samples and has considerable potential for
remote sensing applications (Lopatin et al., 2019; Zhang et al., 2022). Therefore, we selected PLS-SEM to test relatively
important factors and their direct and indirect effects on SP.

When using PLS-SEM with normalized factors, multicollinearity diagnosis, and inner model assessment are essential
for model quality evaluation. Multicollinearity diagnosis (Cenfetelli and Bassellier, 2009), which relies mainly on VIF and
tolerance, is an important issue in model quality evaluation owing to its valuable for unstable factor weights. The primary
assessment of the inner model is $R^2$, which denotes the amount of interpretation variance, and its value often relies on the
research context (Hair et al., 2010). To interpret the SEM results, a standardized path coefficient implies the direct effect of
one factor on another, and its significance ($p < 0.05$) can be evaluated by d resampling procedures (Hair et al., 2011). The path
coefficient was solved iteratively using ordinary least squares, and the significance ($p < 0.05$) of each path coefficient was
obtained by bootstrapping (5000 iterations). The indirect effect of one factor on another can be calculated through multiplying
all indirect path coefficients, and the total effect is the sum of the indirect and direct effects. All the above processes were
implemented by SmartPLS 4 software.

## 4 Results

### 4.1 Accuracy assessment of the extracted snow phenology

Figure 3 shows the accuracy of the satellite-derived SP parameters during 2002–2021, compared against the ground-observed
SP. The satellite-derived SCD, SOD, and SED values were in line with ground-observed ones, with $R$ of 0.77, 0.95, and 0.97
($p < 0.01$), respectively. The bias of ground-observed and satellite-derived SCD, SOD, and SED were −1.75, −2.39, and 3.43
days, indicating that the satellite-derived SCD was less than the actual SCD, and the satellite-derived SOD was earlier than the
actual SOD. In contrast, satellite-derived SED occurred later than the actual SED. Relevant studies have shown the bias of
satellite-derived SP parameters ranges from 0 to 10 days (Chen et al., 2018; Hao et al., 2022; Wang et al., 2021). The biases
(−2.39, 3.43, and −1.75 days) in our study were within this range.





## 4.2 Spatial patterns and temporal trends of snow phenology

The spatial patterns and temporal trends of SP at the pixel level are shown in Figures 4 and 5, respectively. Figure 4a indicates
the spatial distribution of the multiyear averaged SCD from 2002 to 2021, showing a highly uneven snow cover distribution
across the plateau. Areas with an SCD of more than 60 days are considered as stable snow cover areas and sources of water
(Tang et al., 2022), accounting for approximately 17.69% of the total TP and primarily distributed in high-elevation mountain
ranges. Some high-elevation areas had an SCD exceeding 180 days, accounting for approximately 7.55% of the entire TP,
distributed primarily in the western Kunlun, Himalaya, and Nyainqentanglha mountains. Areas of low SCD (< 20 days)
accounted for 56.69% of the total TP, mainly concentrated in low-elevation areas and the hinterland of the plateau. Overall,
the pattern of SCD was in line with topography in visual sense, displaying the regularity of a high SCD in high-elevation
mountains and low SCD in the low-elevation plains.

      We also examined the interannual trends of the SCD and their significance. As Figure 5a shows, the SCD decreased in
64.18% of the TP during the 19-year period, and significantly decreased by 4.29% of the entire area at a mean rate of −2.09
days/year ($p < 0.1$), primarily in the west Kunlun, east-central Himalaya, and south Nyainqentanglha mountains. In contrast,
areas with increased SCD accounted for 35.82% of the TP, increased significantly by 1.84% of the entire area ($p < 0.1$), and
was mainly scattered in the east-central part of the TP. SCD showed a decreasing trend in most areas of the TP.

      SOD and SED distributions over the TP also presented extreme spatial heterogeneity, which was visually consistent with
elevation (Figures 4b and 4c). DOS (day of the snow season) 1 is equivalent to September 1 of the previous year. SOD was
mainly concentrated in DOS 90–180 (December–February), whereas SED was concentrated primarily in DOS 150–240
(February–April), indicating snow on the TP mostly disappeared at the end of April. In high-elevation mountain ranges, SOD
appears earlier in general, whereas SED shows later. But in low-elevation areas such as the hinterland of the TP, SOD starts
later and SED appears earlier. Not only are the variations in SOD and SED spatially complicated, but they also exhibited
significant temporal heterogeneity (Figures 5b and 5c). Areas with delayed SOD accounted for 56.86% of the TP, with the
delay significant in approximately 3.35% of the total area, at a mean rate of 3.64 days/year ($p < 0.1$) (Figure 5b). In contrast,
the area with significantly advanced SOD only accounted for approximately 0.6% of the total area and was sparsely in the
north-western and eastern TP at a mean rate of −3.3 days/year. Consistent with SOD, SED exhibited a delayed trend across
most of the TP. One difference was SED delayed faster over large area in the hinterland of the TP (Figure 5c). Areas with
significantly advanced and delayed SED accounted for only 1.35% and 2.07% of the TP, respectively (Figure 5c).

## 4.3 Direct and indirect effects of important factors on snow phenology

Based on the related literature (Guo et al., 2022; Huang et al., 2020; Kang et al., 2019; Qi et al., 2021), 13 initial associated
factors in five categories were selected: meteorological factors (temperature, precipitation, wind speed, specific humidity, and
shortwave radiation), geographical location (latitude and longitude), topography (elevation, slope, and aspect), vegetation
greenness (NDGI), and atmospheric pollution (BC and AOD). Because the construction of SEM requires prior knowledge,





multiple paths in SEM based on the above 13 factors were investigated, and non-significant and low-coefficient paths should be eliminated theoretically. In this step, the path coefficients of AOD were non-significant for each factor or even for all SP. Therefore, we eliminated AOD and calculated the total effect of the remaining factors on SP (Figure 6). Factors with a small absolute total effect were removed because of their low contribution and limited explanations of SP. An appropriate threshold was selected as the basis for factor elimination. Hair et al. (2011) indicated that there is no uniform standard for selecting

thresholds in different research fields. Considering the rationality of factor selection and referring to relevant literature (Shen et al., 2022; Zhang et al., 2022), we used 0.01 as the threshold for factor elimination. Aspect was eliminated for SCD and SED, and aspect and wind speed was eliminated for SOD (Figure 6). The final SEM models were constructed to discuss the direct and indirect effects of the remaining factors.

   The results of the final SEM and its standardized regression path coefficient (PC) ($p < 0.05$) are shown in Figure 7.

Complex interactions exist between SP and different factors on the TP. Meteorological factors had both direct and indirect effects on SP, whereas vegetation greenness only had a direct impact, and geographical location, topography, and atmospheric pollution only had an indirect effect on SP (Figure 7). Meteorological factors were the dominant factors affecting SP, especially the temperature, which had the strongest effect on SCD, SOD, and SED, with a total effect (TE) of −0.708, 0.371, and −0.562, respectively. The influences of precipitation and specific humidity on SCD and SED were also relatively significant (TC >

0.16). However, their influence on SOD activity was limited (absolute TE value < 0.1). The effect of vegetation greenness on SCD was strongest (PC = −0.126) (Figure 7a), whereas it had little effect on SOD and SED (absolute values of PC < 0.04) (Figures 7b and 7c). Geographical location and topographic conditions affected SP by determining the distribution of climate and vegetation greenness. For example, elevation indirectly affected SCD by assessing the distribution of temperature (PC = −0.990), precipitation (PC = −0.301), and specific humidity (PC = −0.554); hence, the TC at elevation on the SCD was 0.510

(Figure 7a). Therefore, the influence of elevation on SP cannot be ignored and is second only to that of temperature (Figure 7). Slope mainly affected SP by influencing the distribution of precipitation and temperature. Its influence on SCD was the largest (TE = 0.188), followed by SED (TE = 0.147), and SOD (TE = −0.08). The PC of longitudinal and shortwave radiation was relatively large (PC = 0.882). The influence of longitude on all the SP parameters was greater than that of latitude. Although the TE of the BC was not particularly large (TE > 0.02), its influence cannot be ignored.

In summary, temperature was the dominant factor affecting all SP parameters on the TP, followed by elevation. Temperature, NDGI, wind speed, and shortwave radiation negatively affected SCD and SED, whereas other factors positively affected SCD and SED. Temperature and NDGI positively affected SOD, whereas the other factors negatively affected SOD.

**5 Discussion**

Compared to traditional statistical methods, the SEM adopted in this study can reveal the direct and indirect relationships

between SP and associated factors, which is suitable for understanding their mediating effects. Based on the results of this model, the influences of relatively important factors are discussed below.



## 5.1 Response of snow phenology to meteorological factors

Meteorological factors had both direct and indirect effects on SP, but the direct effect was much greater than the indirect effect. Temperature was the dominant meteorological factor, followed by precipitation. A temperature below 0 °C is conducive to the

increase and maintenance of snow cover, resulting in longer SCD, earlier SOD, and later SED (Moran-Tejeda et al., 2013; Scalzitti et al., 2016). Non-negligible effect of temperature on SP has been demonstrated previously (Tang et al., 2022). However, previous studies have shown that precipitation has a low contribution to SP (Guo et al., 2022; Huang et al., 2020; Wang et al., 2021), which is inconsistent with the results of our research. This discrepancy may be caused by the different resolutions of precipitation data. The spatial resolution of the precipitation data we used was 1 km, while the resolution in

previous studies was relatively rough (25 km or data from meteorological stations). The 1 km resolution precipitation data we used reveals greater details of precipitation distribution and provides more accurate precipitation information, thus obtaining a more reliable correlation between precipitation and SP.

To further explore the response mechanisms of SP to temperature/precipitation, their relationship was analyzed at the pixel level and at different temperature/precipitation gradients using Pearson's correlation coefficients (Figures 8 and 9). For

79.79% of the area, SCD was negatively correlated with temperature, and areas with strong negative correlations were primarily in eastern TP (Figure 8a). Between −5℃ and 23℃, the negative correlation between SCD and temperature was significant ($p<0.1$) (Figure 8b). Temperature had both positive (56.63% of the TP) and negative (43.37% of the TP) correlations with SOD activity (Figure 8c). However, the correlations were not significant ($p>0.1$) for all temperature gradients (Figure 8e). Consistent with SCD, SED was negatively correlated with temperature (66.38% of the TP area) (Figure 8e). From −10℃ to

−4℃, the negative correlation between SED and temperature was significant ($p<0.1$). SCD and SED were mainly positively correlated with precipitation, and areas with strong positive correlation were concentrated primarily in the hinterland and southwest of the plateau (Figures 9a and 9e). In contrast, SOD was negatively correlated with precipitation in most areas (Figure 9b).

## 5.2 Topography control on snow phenology

Previous studies have demonstrated the effect of complex topography on SP (Guo et al., 2022; Jain et al., 2008; Ma et al., 2020). Still, they did not reveal the specific action processes (direct or indirect) of various topographic factors. This study calculated the PC between topographic factors and SP using SEM, indicating that topographic conditions indirectly affected SP by first affecting meteorological factors (such as temperature and precipitation). Elevation is the primary factor, due to its vital effect on local microclimates, particularly in mountainous regions. In high-elevation mountains, the temperature and

specific humidity are lower and snowfall is higher, creating favourable conditions for snowfall and maintenance. Owing to strong blocking effect from large mountains, most areas of the hinterland of the TP has relatively scarce snow (Wang et al., 2017). Snow is more likely to slide on a steeper slope, which prevents snow accumulation, whereas a flatter slope is conducive





to snow deposition. As shown in Figure 7, slope had little effect on SOD (compared with SCD and SED), indicating that the onset of snow accumulation was mainly influenced by climate and less by slope.

305       Elevation had a significant effect on SP, but whether the interannual variation in SP on the TP was elevation-dependent was unknown. To investigate this uncertainty, we calculated the interannual variation in SP for each elevational gradient (Figure 10). Almost all the SP parameters showed distinct elevation-dependence: SCD decreased as elevation increased from 1200–5800 m, while SOD and SED advanced from 700–2600 m, and SOD was slightly delayed from 5400–5800 m. At elevations above 5800 m, the elevation dependence was not significant. This may be because the temperature remained below

0 °C at higher elevations, there would likely be no effect on snow cover variability. Thus, elevation dependence was not significant at high elevations. Our conclusions are consistent with those of Ma et al. (2023). In contrast, Ma et al. (2020) showed no elevation dependence of SP on the TP. However, their results were based on meteorological stations concentrated in the eastern part of the TP, with the western TP barely represented, which may have caused the disparity in findings.

      The relative importance of temperature versus precipitation for SP shifts with elevation (You et al., 2020). We

investigated the relationships between SP and temperature/precipitation at different elevations to further examine the effects. As shown in Figure 11, the roles of temperature and precipitation in SP varied along diverse elevation gradients. For the SCD, with an increase in elevation in the 100–3000 m range, the importance of temperature was higher than that of precipitation. However, when the elevation reached 3100 m, the impact of precipitation became more substantial and was greater than that of temperature between 3100–6100 m. Temperature was more important than precipitation at higher elevations (>6200 m).

The correlation between SOD and precipitation was stronger than that with temperature between 1800–4700 m, whereas temperature was more important between 4700–6700 m. However, above 6700 m, SCD and temperature/precipitation correlations became complicated, and neither showed greater importance. Similar to SOD, precipitation had a more significant impact on SED between 1800–4000 m, and the importance of temperature and precipitation was almost equal above 6100 m. One difference was that in the 5100–6100 m range, precipitation was still more critical to SED.

325       Overall, we identified that neither temperature nor precipitation showed consistently higher essential as elevation increased. Our findings differ significantly from previous studies, where researchers generally found an elevation threshold where the importance of each changed. For instance, Moran-Tejeda et al. (2013) discovered a threshold elevation of approximately 1400 m, below which temperature was the primary explanatory variable of snow cover in Switzerland, and above which precipitation was a better predictor. Scalzitti et al. (2016) found a range of threshold elevations (1580–2181 m)

that separated the importance of temperature from that of precipitation in the western United States. The reasons for such difference with our research could be as follows: (1) the higher elevation and different climate conditions of the TP, and (2) the unique changes in temperature/precipitation at different elevations on the TP, especially the greater increase in temperature and the greater decrease in precipitation at high elevations. The confounding influence of warming temperatures and varying precipitation remains a challenge in explaining the observed variations in the SP.





## 5.3 Other factors affecting snow phenology

Geographic location determines the total solar radiation received by the plateau. We further quantified longitudinal/latitudinal zonation of the SP, as shown in Figure 12. A greater longitude (i.e., further east) caused a shorter SCD, later SOD, and earlier SED (Figure 12a). From east to west, SCD decreased by 123 days, SOD was delayed by 77 days, and SED advanced by 48 days. The decreased SCD and advanced SED occurred at low longitudes (73°E–82°E), whereas the other areas showed slightly increased SCD and decreased SED. The SOD showed a delayed trend in almost all longitudes. From low to high latitudes, SCD decreased to an inflection point at 34°–35°and then increased gradually. The SOD and SED advanced to inflection points at 29°–30° and 32°–33°, respectively, and then were steadily delayed (Figure 12b). SCD, SOD, and SED generally showed, respectively, decreasing, delayed, and delayed inter-annual variation trends with increasing latitude. Previous studies have mainly concentrated on the response of SP to latitude, whereas the role of longitude has rarely been discussed (Guo et al., 2022; You et al., 2020). Longitude had a non-negligible effect on SP in this study. Because different longitudinal regions receive different amounts of shortwave radiation (PC = 0.876), temperature (PC = −0.219), precipitation (PC = 0.127), humidity (PC = 0.123), and vegetation greenness (PC = 0.382) (Figure 7), the southeast TP is warm and moist, and the northwest is cold and dry. Therefore, the further west on the TP, the longer the SCD, the earlier the SOD, and the later the SED (Figure 12a).

Vegetation greenness can change snow redistribution by intercepting it, and can affect spatial pattern and melt rate of surface snow by regulating solar radiation budget balance. Vice versa, the processes and patterns of snowmelt and snowmelt accumulation affect vegetation greenness, species distribution, and community structure (Barrere et al., 2018; Walker et al., 1993). Most previous studies explored the response of vegetation greenness to snow cover, and the role of vegetation greenness on snow cover has rarely been investigated (Qi et al., 2021; Xu et al., 2022b). Sturm et al. (2001) reported that vegetation greenness favours the aggregation of wind-drifted snow, forming snow layers with lower thermal conductivity, thereby increasing the snowpack insulation capacity. Domine et al. (2016) found that by intercepting drifting snow, vegetation greenness increased snow height and reduced snow density and thermal conductivity. These studies suggest that vegetation greenness enhances snow accumulation with lower thermal conductivity. In our study, vegetation greenness was positively correlated with SOD and negatively correlated with SCD and SED. However, its direct and indirect effects were minimal (TE = 0.028, 0.126, and 0.037, respectively) (Figure 7). Vegetation greenness was not the main factor affecting SP.

## 6 Conclusions

In this study, based on the daily gap-free HMRFS-TP dataset, we investigated the spatiotemporal variation of SP on the TP from 2002 to 2021. We quantified the direct and indirect effects of the associated factors on SP. The spatial patterns of SP were vertically zonal, with higher elevations having a longer SCD, earlier SOD, and later SED. The interannual trends of all SP parameters varied in different zones (e.g., elevation and longitude) and contributed to the general trend in variation between 2002–2021: decreased SCD, delayed SOD, and slightly delayed SED. In particular, an elevation-dependent pattern of interannual variation in SP was observed below 5800 m. The direct and indirect effects of various associated factors on SP





were analyzed in detail, among which meteorological factors had both direct and indirect effects; vegetation greenness had only a direct impact; and geographical location, topography, and atmospheric pollution had only indirect effects on SP. Undoubtedly, meteorological factors were the absolute dominant factors in particular temperature. However, the influences of

topography, geographic location, vegetation greenness, and atmospheric pollution cannot be ignored. As two rather important factors, we identified that neither temperature nor precipitation showed consistently high importance as elevation increased.

This study explored the dynamic variation in snow cover and revealed the mediating effects of multiple factors in its changing process, which contributed to providing a strategic basis for predicting and solving the problems of climate change, hydrological cycle, and ecological balance in the future in the context of global warming on the TP.

*Author contributions.* JHX and YH designed and performed the experiments. YT and LXD performed the extraction of snow phenology. SJW, BLY, and JPW processed the validation experiments. ZJZ performed data curation. All the authors contributed to the analysis and writing of this paper.

*Competing interests.* The contact author has declared that neither they nor their co-authors have any competing interests.

*Acknowledgements.* This work was supported by the National Natural Science Foundation of China (no. 42071306). We would

like to thank the National Meteorological Centre of China for providing the *in situ* observations over the Tibetan Plateau.

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



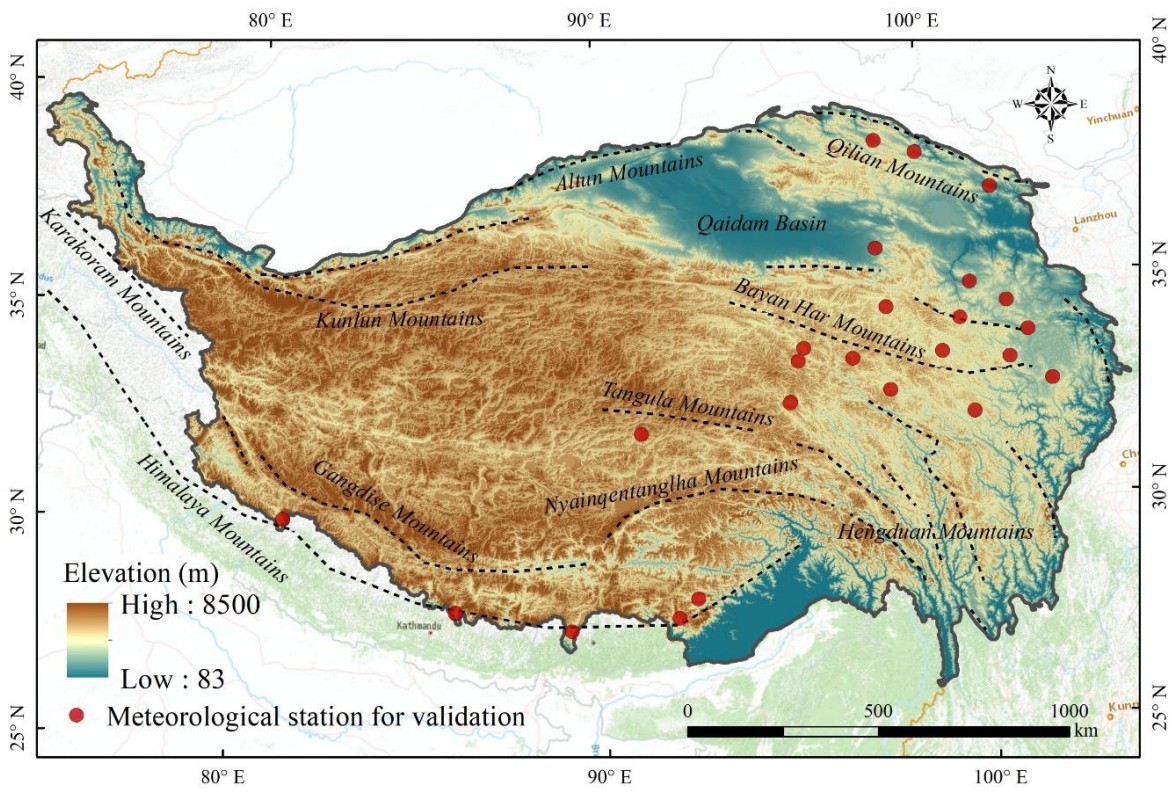

**Figure 1: Topography and distribution of selected meteorological stations on the Tibetan Plateau (TP) used for data validation (base map from ESRI).**





**Figure 2: Flowchart to reveal the spatiotemporal variability of snow phenology (SP) and the direct and indirect effects of associated factors on the TP from 2002 to 2021. SEM refers to the Structural Equation Model.**





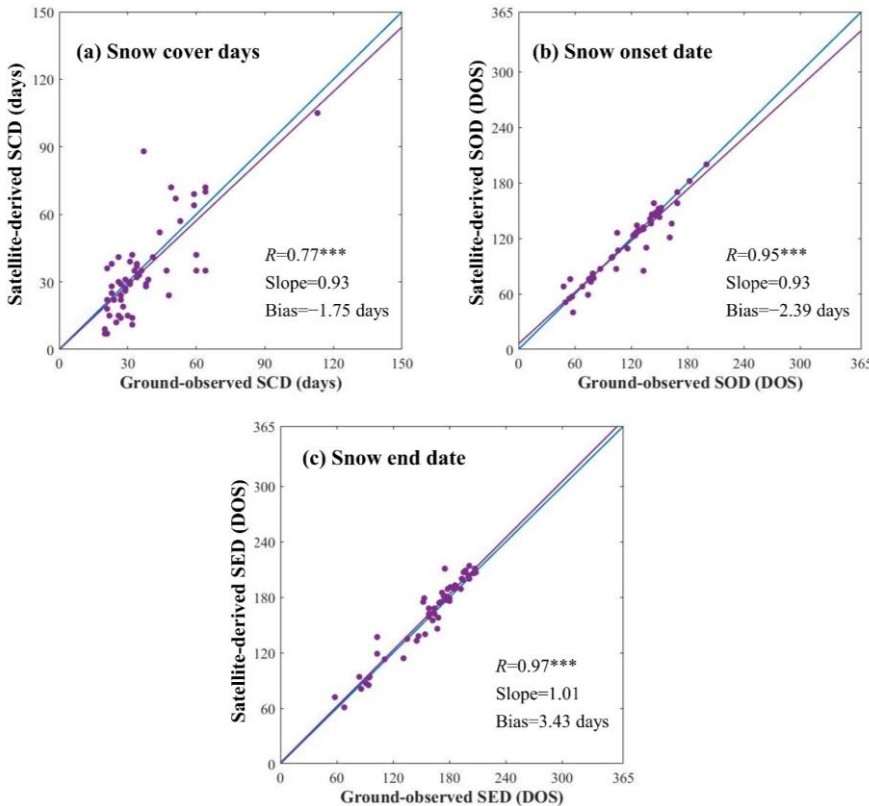

**Figure 3: The accuracy of SP parameters evaluated by ground-observed values from 2002 to 2021. (a) SCD, (b) SOD, and (c) SED.**
**Note: *** indicates significance at the level of 0.01. DOS represents the day of the snow season.**



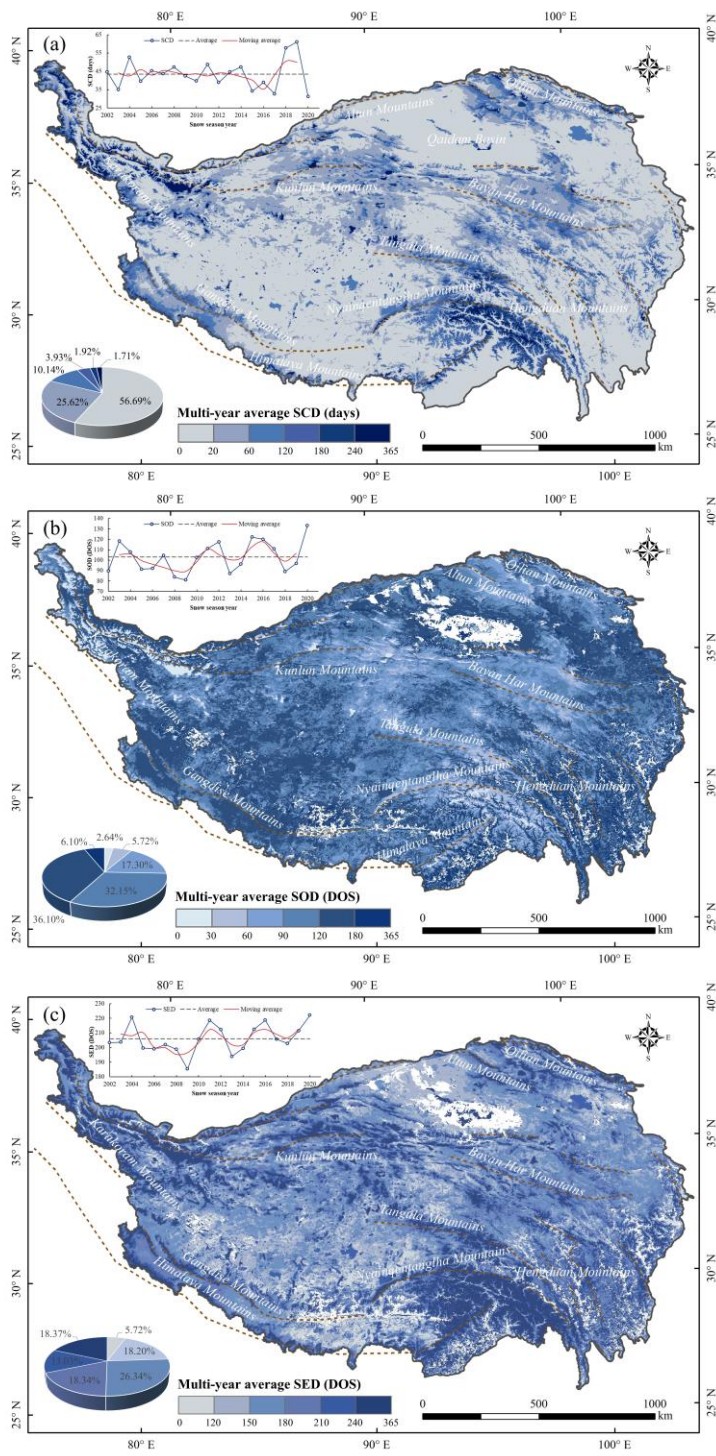

**Figure 4: The spatial pattern of the multiyear averaged SP on the TP from 2002 to 2021. (a) SCD, (b) SOD, and (c) SED. Note: pixels which have not been detected in 19 years are regarded as snow-free pixels (i.e., the white area in the figure).**



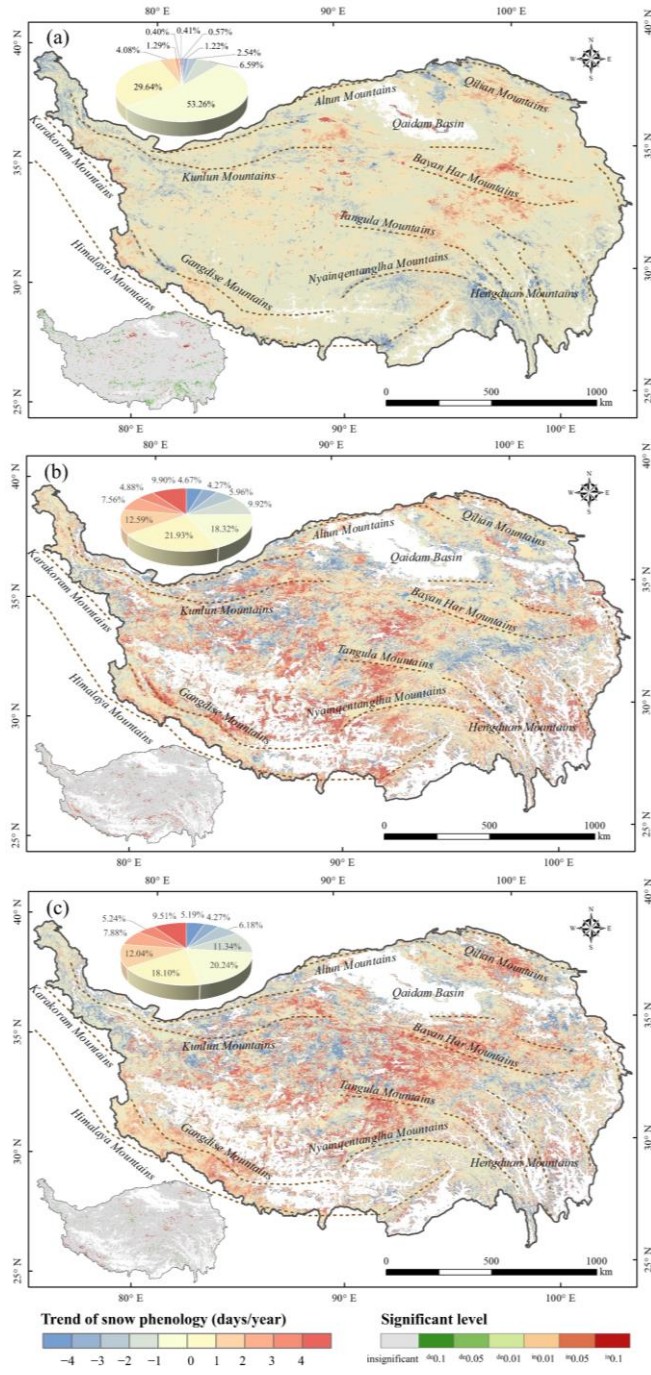

**Figure 5: Interannual variation trend of the SP from 2002 to 2021. (a) SCD, (b) SOD, and (c) SED. Note: the map at the bottom left of each subgraph indicates the significant level. The symbol "de" in the legend indicates a significantly decreased trend and "in" indicates a significantly increased trend.**





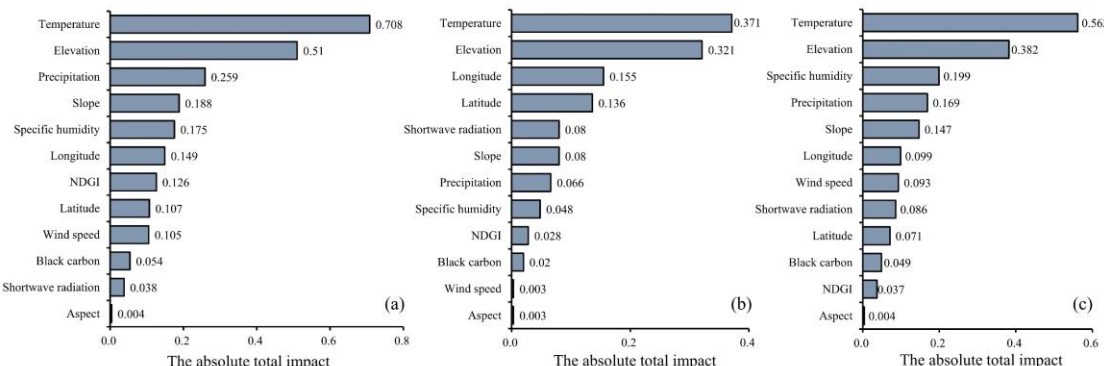

Figure 6: The absolute total effect of factors affecting SP based on SEM. (a) SCD, (b) SOD, and (c) SED.



**Figure 7: The final SEM of each SP parameter. (a) SCD, (b) SOD, and (c) SED. Note: the red line implies a positive effect, while the blue line denotes a negative effect. All path coefficients are statistically significant ($p < 0.05$).**





**Figure 8: Correlation between temperature and SP at pixel level of (a) SCD, (c) SOD, and (e) SED; and at different temperature gradients for (b) SCD, (d) SOD, and (f) SED.**







**Figure 9: Correlation between precipitation and SP at pixel level of (a) SCD, (c) SOD, and (e) SED; and at different precipitation gradients for (b) SCD, (d) SOD, and (f) SED.**



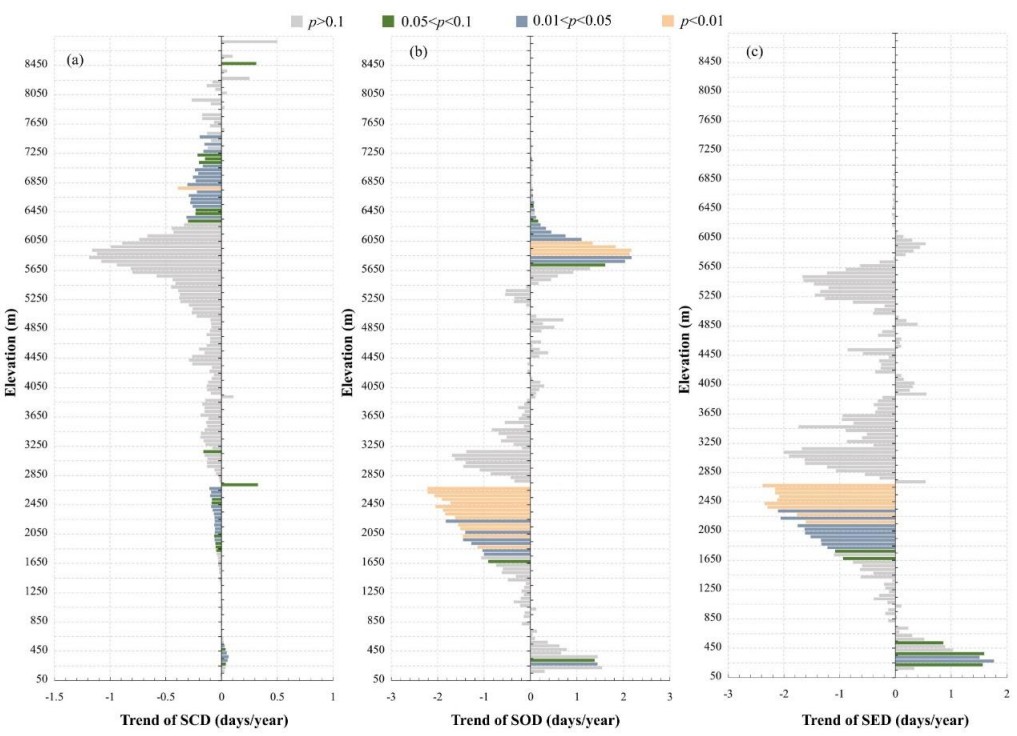

**Figure 10: Interannual variation trend of SP under different elevation gradients from 2002 to 2021. (a) SCD, (b) SOD, and (c) SED.**

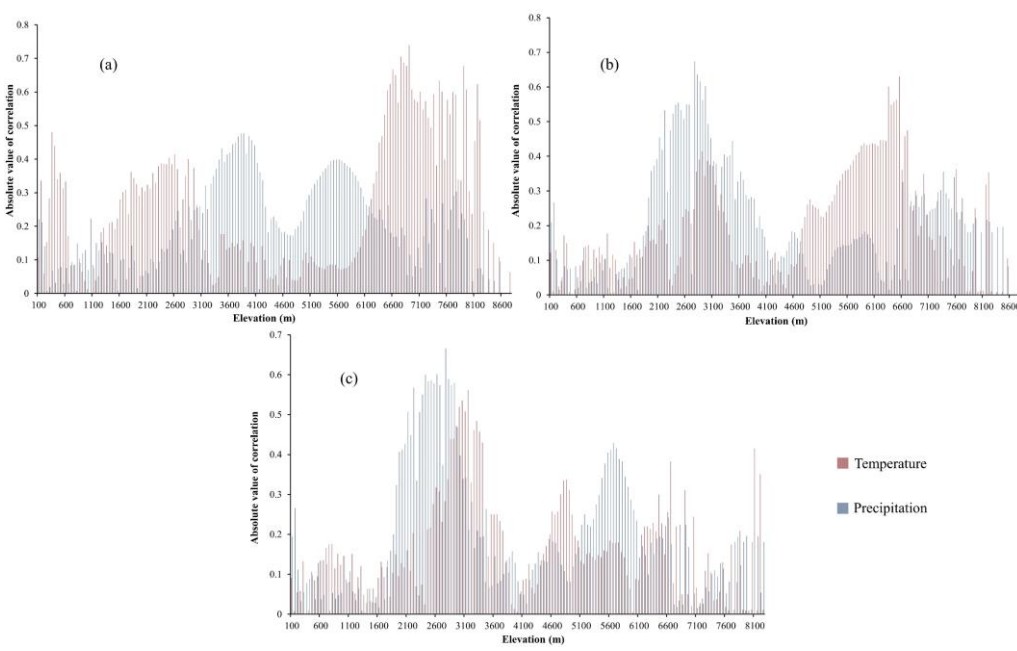

**Figure 11: The correlation between SP and temperature/precipitation under different elevation gradients. (a) SCD, (b) SOD, and (c) SED.**



**Figure 12: Multiyear averaged SP and interannual variation trends associated to (a) longitude and (b) latitude. Note: no samples at longitude 78°E−79°E.**