# Peer review of "Temperature-dominated spatiotemporal variability in snow phenology on the Tibetan Plateau from 2002 to 2022"

_The Cryosphere, 2023_

## Author Comment (AC3)

**Author's response**

The authors investigated the spatial-temporal changes in snow phenology using a high-resolution snow cover data. This study is very important for understanding the response of cryosphere to changing climate. Overall, the manuscript is well written and presented and I believe it is a good bit of work with a clear conclusion reached. I have two minor concerns:

Figure 10: First, I am not sure whether those trends with p>0.05 could still be used in such an analysis since a statistically in-significant trends are usually treated as "no change". Even we incorporate them, the controls of elevation on the SP seem to be valid only at certain elevation range. Taking Figure 10a as an example, while it is indeed clear that trends in SCP depends on elevation from ~5000 to 5800 m, the dependence becomes unclear from 50 to ~5000 m (At least I cannot see a clear pattern of "SCD decreased as elevation increased from 1200–5800 m" from 10a). The authors further stated that "At elevations above 5800 m, the elevation dependence was not significant", but Figure 10b yields a clear elevation dependence for SOD. Therefore, it may better to describe more details (e.g., the specific elevation range information) about the elevation dependence in Section 5.2. In addition, can any quantitative analysis (e.g., the correlation between the trend and elevation?) be added for such an analysis? I would say elevation dependence is true if the correlation is statically significant, otherwise not. Lastly, it would be also nice if you can tell the detailed rule of the elevation dependence, e.g., the changes in trend for every 100 m rise in elevation. By the way, for Figure 10 caption, I would suggest just using "Trends in SP……" as a start here. There is no need to say "interannual variation trend". Please revise this issue also in the main text.

**Response:** Thank you for the valuable suggestions, which have helped us signficiantly improve the manuscript. We agree that statistically insignificant trends are not suitable for further analysis between trends in SP and elevation. In this revision, we divided elevation into 50 m intervals, and the average trend values of statistically significant pixels ($p < 0.1$) were calculated for each elevation category. Then, we selected elevation categories with more than 50 samples to analyze the correlation between trends in SP and elevation (Figure 1).

Our finding reveals a strong negative correlation between trend in SCD and elevation, with a correlation coefficient of −0.73 (Figure 1a). This negative correlation was strongest in 0−4000 m and 4900−5900 m. A moderate positive correlation exists between the trend in SOD and elevation ($R = 0.59$), and this positive correlation was most significant in 4100−5800 m (Figure 1b). The correlation between SED and elevation was 0.33, which was not significant (Figure 1c). Therefore, we conclude that there exists a strong elevation dependence for the trend in SCD ($R = −0.73$), a moderate elevation dependence for the trend in SOD ($R = 0.59$), and no significant elevation dependence for the trend in SED ($R = 0.33$) from 2002 to 2022 on the TP.

[Figure]

**Figure 1:** Scatter plot of elevation and trends in snow cover days (a), snow onset date (b), and snow end date (c) from 2002 to 2022.

The caption of Figure 10 has been changed as "Trends in SP……" , also for the main text. Related discussion has been added into *Discussion 5.2*:

*To investigate the elevation effect, we divided elevation into 50 m intervals, and the average trend values of statistically significant pixels (p < 0.1) were calculated for each elevation category. Then, we selected elevation categories with more than 50 samples to analyze the correlation between trends in SP and elevation. Our finding reveals a strong negative correlation between trend in SCD and elevation, with a correlation coefficient of −0.73 (Figure 9a). This negative correlation was strongest in 0−4000 m and 4900−5900 m. A moderate positive correlation exists between the trend in SOD and elevation (R = 0.59), and this positive correlation was most significant in 4100−5800 m (Figure 9b). The correlation between SED and elevation was 0.33, which was not significant (Figure 9c). Therefore, there exists a strong elevation dependence for the trend in SCD (R = −0.73), a moderate elevation dependence for the trend in SOD (R = 0.59), and no significant elevation dependence for the trend in SED (R = 0.33) from 2002 to 2022.*

Given a warming background, it is easy to understand why SOD delayed, but can you please explain why the SED delayed across most of TP (Line 233)?

**Response:** In this revision, we developed a structural equation model to explain the causes of the delayed trend in SED (Figure 2). The analysis shows that wind speed significantly influences delayed trend in SED, showing a strong positive correlation with a total effect of 0.497 (Figure 2). The delayed trend in SED becomes more pronounced as wind speed increases. This effect of wind speed on snow cover is mainly through blowing snow process, which lead to increased snow accumulation through snow redistribution and thus a delayed trend in SED regionally. Li et al. (2012) also found a distinct occurrence of blowing snow at an elevation of 4146 m, where pronounced redistribution of snow cover occurred.

[Figure]

**Figure 2:** The structural equation model based on delayed trend in SED and influencing factors. Note: the red line implies a positive effect, while the blue line denotes a negative effect. All path coefficients are statistically significant ($p < 0.05$).

Therefore, we hypothesize that the delayed trend of SED may be due to the redistribution of snow cover caused by blowing snow. However, future studies are necessary to look into the distinctive impacts that blowing snow has on the processes of snow accumulation and subsequent melt. We have added the following content in *Discussion 5.2*:

*The non-significant elevation dependence of SED is due to the competing effects of the delayed trend within 3800−4900 m elevation and the advanced trend of other elevation ranges (Figure 9c). To explain the counterintuitive delayed trend in the context of regional warming, we further applied a structural equation model to explore the causal relationships. The results indicate that wind speed significantly influences the delayed trend in SED, exhibiting a strong positive correlation with a total effect of 0.497 (Figure 9d). The delayed trend in SED becomes more pronounced as wind speed increases. This effect of wind speed on snow cover is mainly through blowing snow process, which lead to increased snow accumulation through snow redistribution and thus a delayed trend in SED regionally. Li et al. (2012) also found a distinct occurrence of blowing snow at an elevation of 4146 m, where pronounced redistribution of snow cover occurred.*

**References:**

Li, H., Wang, J., Hao, X.: Influence of Blowing Snow on Snow Mass and Energy Exchanges in the Qilian Mountainous, Journal of Glaciology and Geocryology, 34(05),1084-1090, 2012. (in Chinese)

---

## Author Response (AR1)

**Response to Reviewer #1**

**General comments:**

This paper analyzes the spatiotemporal variations in snow phenology on the Tibetan Plateau and elucidates the comprehensive effects of various influencing factors. It provides crucial insights for addressing climate change, hydrological cycles, and ecological balance issues in the context of global warming. The experimental design is well-structured and the result is complete. However, several issues still need to be clarified before being published in *The Cryosphere*. Further consideration is necessary regarding the selection of influencing factors, as well as the resolution and time delay effects of meteorological factors. Details and suggestions are given below.

**Response:** We sincerely appreciate your constructive feedback and have revised our manuscript accordingly. In this revision, we have used a new meteorological dataset with 3 km resolution to reconstruct the Structural Equation Model (SEM), and expanded the discussion on the time-delay effects of these meteorological factors. We have also discussed the limitations of the study, such as the effect of black carbon and other important factors that have not been considered before. The specific responses to each of the suggestions as follows.

**Special comments**

**Comment 1:**
Section 2.1: it is essential to add detailed information on the spatiotemporal distribution of snow cover in the Tibetan Plateau, as well as temperature and precipitation patterns.
**Response:** As suggested, we have added the following sentences in Section 2.1:

*The TP (26°00'12"N-39°46'50"N, 73°18'52"E-104°46'59"E, Figure 1) is the largest snow cover area in the middle latitudes of the Northern Hemisphere, with $10^5$ km² of glaciers and an annual snowfall of $41.9 \times 10^9$ m³ (Yao et al., 2012). The southeastern TP is warm and moist while the northwestern TP is cold and dry. The average annual temperature ranges from −6 to 20 ℃ and the average annual precipitation ranges from 150 to 800 mm (Wang et al., 2023).*

**References:**
Yao, T., Thompson, L. G., Mosbrugger, V., Zhang, F., Ma, Y., Luo, T., Xu, B., Yang, X., Joswiak, D. R., Wang, W., Joswiak, M. E., Devkota, L. P., Tayal, S., Jilani, R., and Fayziev, R.: Third Pole Environment (TPE), Environmental Development, 3, 52-64, 10.1016/j.envdev.2012.04.002, 2012.

Wang, Z., Huang, L., and Shao, M. a.: Spatial variations and influencing factors of soil organic carbon under different land use types in the alpine region of Qinghai-Tibet Plateau, Catena, 220, https://doi.org/10.1016/j.catena.2022.106706, 2023.

**Comment 2:**

Section 2.2: clearly states that spatiotemporal analysis in SP extends until 2021, but due to limitations in the data for influencing factors, the analysis is based on data up to 2018.

**Response:** In this revision, the analysis has been extended to 2022 using the new meteorological dataset (2002−2022).

**Comment 3:**

The accumulation and melting of snow can be influenced by various factors. In addition to the 13 factors mentioned in the manuscript, land surface temperature, soil characteristics (such as soil moisture), frozen soil, human activities, and other relevant factors can also affect snow cover variation. It is suggested to provide the analysis for these factors.

**Response:** Indeed, various factors can affect the accumulation and melting of snow, such as land surface temperature, soil moisture, frozen soil, human activities, and other relevant factors. Ground temperature primarily influences the structure and stability of snowpack by regulating energy exchange at the soil-snow interface (Rixin et al., 2022). As the ground temperature increases, the substrate absorbs additional thermal energy, which is conveyed to the base of the accumulated snow through heat conduction, resulting in melting of the lower snow layers. The heat from the warmer soil in the snow-free area can be transferred to the colder soil below the snow-covered area. Liquid water can also be transferred from the snow-free soil to the snow-covered soil, thus melting snow (Fassnacht et al., 2006). Therefore, soil properties (e.g., soil moisture) can also affect snow cover. Despite the potential importance of these factors to the SP of TP, they were not analyzed in this study due to limited data availability of these factors over extended spatial and temporal scales. Due to the sparse population and minimal human development, human activities are typically not considered in snow cover studies on the TP. In this revision, we have discussed the importance of the aforementioned factors in *Discussion 5.3*:

*The dynamics of snow accumulation and melting are influenced by various factors. In addition to the factors analyzed in this paper, other factors may also play important roles in SP. Ground temperature primarily influences the structure and stability of snowpack by regulating energy exchange at the soil-snow interface (Rixin et al., 2022). As the ground temperature increases, the substrate absorbs additional thermal energy, which is conveyed to the base of the accumulated snow through heat conduction, resulting in melting of the lower snow layers. The heat from the warmer soil in the snow-free area can be transferred to the colder soil below the snow-covered area. Liquid water can also be transferred from the snow-free soil to the snow-covered soil, thus melting snow (Fassnacht et al., 2006). Therefore, soil properties (e.g., soil moisture) can also affect snow cover. Despite the*

*potential importance of these factors to the SP of TP, they were not analyzed in this study due to limited data availability of these factors over extended spatial and temporal scales. Developing high-resolution, spatiotemporal continuous datasets for these factors will be useful in future efforts to comprehensively quantify the response of SP to changing climate conditions.*

**References:**

Fassnacht, S. R., Yang Z. L., Snelgrove, K. R., Soulis, E. D., and Kouwen, N.: Effects of Averaging and Separating Soil Moisture and Temperature in the Presence of Snow Cover in a SVAT and Hydrological Model for a Southern Ontario, Canada, Watershed, Journal of Hydrometeorology, 7, 298–304, https://doi.org/10.1175/JHM489.1, 2006.

Rixen, C., Høye, T. T., Macek, P. et al.: Winters are changing: snow effects on Arctic and alpine tundra ecosystems, Arctic Science, 8, 572-608, 10.1139/as-2020-0058, 2022.

**Comment 4:**

As far as I know, meteorological data with a spatial resolution of approximately 3 km is now available for the Tibetan Plateau (https://data.tpdc.ac.cn/zh-hans/data/44a449ce-e660-44c3-bbf2-31ef7d716ec7). It is suggested to use this data to analyze the impact of meteorology.

**Response:** As suggested, we have used this meteorological data with a spatial resolution of 3 km in our revision. We have added a description to this dataset in the *Section 2.2.2*:

*We used daily meteorological dataset at 1/30° resolution from 2002 to 2022 to investigate the impact of meteorological factors on SP, including air temperature, precipitation, specific humidity, wind speed, and downward shortwave radiation. This dataset was generated by integrating in situ observations, remote sensing, and reanalysis dataset (He et al., 2020; Yang et al., 2023). All daily meteorological data were resampled to 500 m to maintain consistency with the MODIS snow product. We calculated average air temperature, average humidity, average wind speed, total precipitation, and total shortwave radiation for each snow season, snow accumulation season, and snowmelt season to examine their relationship with SCD, SOD, and SED, respectively (Chen et al., 2018; Ma et al., 2023).*

We have modified the results in the *Section 4.3*:

*Figure 7 presents the results of the final SEM and its standardized regression path coefficient (PC) (p < 0.05). It indicates that there are complex interactions between SP and associated factors on the TP. Temperature, wind speed, NDGI, and shortwave radiation had negative effects on SCD and positive effects on SOD, while the other factors had positive effects on SCD and negative effects on SOD. Furthermore, temperature, wind speed, and shortwave radiation had negative effects on SED, while the remaining factors had positive effects on SED. Meteorological factors (air temperature, precipitation, wind*

*speed, and shortwave radiation) exerted both direct and indirect effects on SP, whereas vegetation greenness (NDGI) mainly had a direct impact, and geographical location (latitude and longitude) and topography (elevation and slope) had an indirect effect on SP (Figure 7). Meteorological factors were the dominant factors affecting SP, especially the temperature, which had the strongest effect on SCD, SOD, and SED, with the TE value of −0.447, 0.272, and −0.379, respectively. The influences of precipitation on SCD and SOD were also relatively significant (absolute value of TE > 0.19). However, its impact on SED was limited (TE=0.049). Wind speed exhibited a strong effect on SCD and SED (absolute value of TE > 0.15), while its effect on SOD was relatively limited (TE=0.062). The effect of NDGI on SCD and SOD was strongest (absolute value of TE > 0.17). Geographical location and topographic conditions affected SP by determining meteorological conditions and the growth of vegetation. For example, elevation indirectly affected SCD by assessing the distribution of temperature (PC = −0.805), precipitation (PC = −0.261), shortwave radiation (PC = 0.330), wind speed (PC = 0.418), and NDGI (PC = −0.649); hence, the TE of elevation on the SCD was 0.290 (Figure 7a). The influence of latitude on all SP parameters was greater than that of longitude.*

[Figure]

**Figure 7: The SEM of each SP parameter of (a) SCD, (b) SOD, and (c) SED. Note: the red line implies a positive effect, while the blue line denotes a negative effect. All path coefficients are statistically significant (p < 0.05).**

**Comment 5:**

Meteorological data can exhibit a time delay effect on SP. For instance, temperatures may rise for several days before the actual snowmelt process starts. Consequently, relying solely on the annual average temperature for analysis is not entirely reasonable. Please try to explain this delay effect in the given context.

**Response:** There indeed exists a time delay effect when considering meteorological factors. In fact, apart from temperature, other meteorological factors may also have a lag effect on snow cover. It is also important to note that the duration of the delay effect may vary across different meteorological factors. For instance, anomalous precipitation leads to subsequent snow cover variations in the TP with a delay of approximately 5 days (Li et al., 2019). Ren et al. (2018) observed a strong negative correlation between snow cover and the average temperature of 1 to 4 months prior to snow accumulation. Since the structural equation model investigates the influences between various factors, it is imperative to ensure that the study time periods for all factors are synchronized. This consistency allows for a meaningful comparison and accurate assessment of the causal relationships within the model. Consequently, this paper does not delve into the time delay effect of meteorological factors. In our revision, we discussed the time delay effect in *Discussion 5.1*:

*It should be noted that our study did not account for the lagging effect of meteorological factors on snow cover. Previous studies suggested potential lagging effects of meteorological factors on the SP of TP. For instance, Li et al. (2019) found that anomalous precipitation could lead to subsequent snow cover variations on the TP with a delay of approximately 5 days. Ren et al. (2018) observed a strong negative correlation between snow cover and the average temperature of 1 to 4 months prior to snow accumulation. Despite the potential importance of lagging effects on the snow cover of TP, our SEM model requires concurrent observations of dependent and independence variables for accurate quantification of the causal relationships between them. Therefore, the lagging effects were neglected in this study.*

**References:**

Li, W., Qiu, B., Guo, W., Zhu, Z., and Hsu, P. C.: Intraseasonal variability of Tibetan Plateau snow cover, International Journal of Climatology, 40, 3451-3466, https://doi.org/10.1002/joc.6407, 2019.
Ren, Y., Liu, S.: Different influences of temperature on snow cover and sea ice area in the Northern Hemisphere, Geographical Research, 37(05), 870-882, 2018. (in Chinese)

**Comment 6:**

The authors have considered the impact of black carbon and similar factors; however, their influence does not seem significant. Please explain the reasons behind this, especially in light of previous studies indicating the importance of black carbon.

**Response:** The global black carbon emission dataset we previously utilized focuses on the emission of black carbon (Xu et al, 2021), which may be not suitable for representing the transportation and deposition of black carbon on the Tibetan Plateau.

Previous research has mainly relied on numerical simulations from climate models to investigate the effect of black carbon on snow cover on the TP, such as RegCM4.3 (Ji et al., 2015) and GEOS-5 (Lau et al., 2018). Furthermore, analysis of back trajectory methods also serves as a vital approach to understanding the significance of black carbon on snow cover (Zhang et al., 2018). However, these methods do not generate a black carbon dataset that can be directly utilized as input for structural equation model. To our knowledge, the MERRA-2 reanalysis dataset is the most widely used black carbon concentrations product (Chen et al., 2023; Xu et al., 2020). Its spatial resolution ($0.5° \times 0.625°$, approximately 50 km $\times$ 60 km) renders it more appropriate for large-scale investigations (such as Arctic or China), which is not suitable for the Tibetan Plateau. In our latest revision, we have removed the experimental section about black carbon, and have added the following content to the *Discussion 5.3*:

*Atmospheric pollutants, especially those referred to as light-absorbing aerosols, such as black carbon, brown carbon and dust, can warm the atmosphere (Kang et al., 2019; Ji et al., 2015). After being deposited onto snowpack, these light-absorbing particles can reduce the surface albedos of snowpack and promote its melting (Zhang et al., 2018; Lau et al., 2018). Despite the potential importance of these factors to the SP of TP, they were not analyzed in this study due to limited data availability of these factors over extended spatial and temporal scales. Developing high-resolution, spatiotemporal continuous datasets for these factors will be useful in future efforts to comprehensively quantify the response of SP to changing climate conditions.*

**References:**

Ji, Z., Kang, S., Cong, Z., Zhang, Q., Yao, T.: Simulation of carbonaceous aerosols over the third pole and adjacent regions: distribution, transportation, deposition, and climatic effects, Climate Dynamics, 45 (9), 2831–2846, https://doi.org/10.1007/s00382-015-2509-1, 2015.

Kang, S., Zhang, Q., Qian, Y., Ji, Z., Li, C., Cong, Z., Zhang, Y., Guo, J., Du, W., Huang, J., You, Q., Panday, A. K., Rupakheti, M., Chen, D., Gustafsson, O., Thiemens, M. H., and Qin, D.: Linking atmospheric pollution to cryospheric change in the Third Pole region: current progress and future prospects, National Science Review, 6, 796-809, https://doi.org/10.1093/nsr/nwz031, 2019.

Lau, W. and Kim, K.-M.: Impact of Snow Darkening by Deposition of Light-Absorbing Aerosols on Snow Cover in the Himalayas–Tibetan Plateau and Influence on the Asian Summer Monsoon: A Possible Mechanism for the Blanford Hypothesis, Atmosphere, 9, https://doi.org/10.1007/10.3390/atmos9110438, 2018.

Xu, X., Yang, X., Zhu, B., Tang, Z., Wu, H., and Xie, L.: Characteristics of MERRA-2 black carbon variation in east China during 2000–2016, Atmospheric Environment, 222, https://doi.org/10.1016/j.atmosenv.2019.117140, 2020.

Zhang, Y., Kang, S., Sprenger, M., Cong, Z., Gao, T., Li, C., Tao, S., Li, X., Zhong, X., Xu, M., Meng, W., Neupane, B., Qin, X., and Sillanpää, M.: Black carbon and mineral dust in snow cover on the Tibetan Plateau, The Cryosphere, 12, 413-431, https://doi.org/10.5194/tc-12-413-2018, 2018.

**Comment 7:**
It is suggested to discuss the limitations of the study.

**Response:** We have discussed the limitations of time delay effects of meteorological factors, as well as the effect of some important factors that have not been considered before.

*5.1 Response of snow phenology to meteorological factors*

*This study still has some limitations in exploring the effect of meteorological factors on SP. We calculated the average (or total) metrics of each meteorological factor during the snow accumulation season to explore their relationship with SOD using SEM. Similarly, we calculated the average (or total) metrics of each meteorological factor during the snowmelt season to investigate their relationship with SED (Chen et al., 2018; Ma et al., 2023). The purpose of this is to address the time dependence of meteorological factors. In the future, we intend to explore more methods for effectively separating the time dependence of meteorological factors when studying their effects on snow phenology. Additionally, previous studies suggested potential lagging effects of meteorological factors on the SP of TP. For instance, Li et al. (2019) found that anomalous precipitation could lead to subsequent snow cover variations on the TP with a delay of approximately 5 days. Ren et al. (2018) observed a strong negative correlation between snow cover and the average temperature of 1 to 4 months prior to snow accumulation. Despite the potential importance of lagging effects on the snow cover of TP, our SEM model requires concurrent observations of dependent and independence variables for accurate quantification of the causal relationships between them. Therefore, the lagging effects were neglected in this study.*

*5.3 Other factors affecting snow phenology*

*The dynamics of snow accumulation and melting are influenced by various factors. In addition to the factors analyzed in this paper, other factors may also play important roles in SP. Ground temperature primarily influences the structure and stability of snowpack by regulating energy exchange at the soil-snow interface (Rixin et al., 2022). As the ground temperature increases, the substrate absorbs additional thermal energy, which is conveyed to the base of the accumulated snow through heat conduction, resulting in melting of the lower snow layers. The heat from the warmer soil in the snow-free area can be transferred to the colder soil below the snow-covered area. Liquid water can also be transferred from the snow-free soil to the snow-covered soil, thus melting snow (Fassnacht et al., 2006). Therefore, soil properties (e.g., soil moisture) can also affect snow cover. In addition, atmospheric pollutants, especially those referred to as light-absorbing aerosols, such as black carbon, brown carbon and dust, can warm the atmosphere (Kang et al., 2019; Ji et al., 2015). After being deposited onto snowpack, these light-absorbing particles can reduce*

*the surface albedos of snowpack and promote its melting (Zhang et al., 2018; Lau et al., 2018). Despite the potential importance of these factors to the SP of TP, they were not analyzed in this study due to limited data availability of these factors over extended spatial and temporal scales. Developing high-resolution, spatiotemporal continuous datasets for these factors will be useful in future efforts to comprehensively quantify the response of SP to changing climate conditions.*

**Minor revisions:**

1. Line 113: There is a missing space between "2002" and "to."
**Response:** We have added a space here.

2. Line 194: There is an extra "d" between "by" and "resampling."
**Response:** We have removed the extra word.

3. Line 234: There is a missing space between "2.07%" and "of."
**Response:** We have added a space here.

4. The slope labels for Figure 3a and Figure 3b are the same, but they appear to be inconsistent. Please double-check.
**Response:** We have corrected the slope label in Figure 3a to 0.96.

[Figure]

**Figure 3:** The accuracy of SP parameters evaluated by ground-observed values from 2002 to 2021. (a) SCD, (b) SOD, and (c) SED. Note: *** indicates significance at the level of 0.01. DOS represents the day of the snow season.

**Response to Reviewer #2**

In this paper, the authors examine the spatiotemporal variability of the snow phenology over the Tibetan Plateau, based on MODIS satellite observations. The period covers 2002 to 2021.

The study focuses on snow onset, snow end date and snow cover days. It relies on an improved MODIS dataset, filled for temporal and spatial data gaps. Governing factors, which can have both direct and indirect effects on the snow phenology through the interplay of the different variables are extracted by a Structure Model. It is found that the meterorological factors (temperature, precipitation) play the leading roles. However, the relative importance of temperature and precipitation shifts with elevation.

The study is detailed and comprehensive, and the article clearly written. It should prove a valuable to understand the snow cover variability over the Tibetan Plateau. I recommend the paper for publication provided the three main comments (essentially, discussion points) are addressed.

**Response:** We appreciate your valuable suggestions and comments, which have contributed to substantial improvement of the manuscript. As suggested, we have particularly improved the manuscript in the discussion section, where we discussed more detail about the effect of black carbon on snow cover, time-dependency of meteorological factors, and the complex relationship between elevation and longitude on snow cover. Detailed responses to each of the comments are provided below.

**Main comments:**

**Comment 1:**
It is a bit surprising that the "darkening of the snow" plays such a minor role, given the attention given to this issue in recent years. [e.g., W. Lau et al, Atmosphere 2018, 9(11),https://doi.org/10.3390/atmos9110438]. Are the findings consistent with earlier studies about this point? Is the global BC emission dataset really relevant here since it is mostly the BC transported and deposited on the Tibetan Plateau that would matter? Is the AOD dataset of sufficient quality and resolution over the area of interest?

**Response:** We agree that the global black carbon emission dataset we previously utilized focuses on the emission of black carbon (Xu et al, 2021), which may be not suitable for representing the transportation and deposition of black carbon on the Tibetan Plateau. The AOD data we previously used measures the aggregate concentration of atmospheric particulate matter, but fails to differentiate black carbon from other particulate components, such as PM2.5 or PM10 (Bai et al., 2022). Therefore, our previous analysis that relied on the global black carbon emission dataset and AOD may exhibit certain biases.

Previous research has mainly relied on numerical simulations from climate models to investigate the effect of black carbon on snow cover on the TP, such as RegCM4.3 (Ji et al., 2015) and GEOS-5 (Lau et al., 2018, reference mentioned by referee). Furthermore, analysis of back trajectory methods also serves as a vital approach to understanding the significance of black carbon on snow cover (Zhang et al., 2018). However, these methods do not generate a black carbon dataset that can be directly utilized as input for structural equation model. To our knowledge, the MERRA-2 reanalysis dataset is the most widely used black carbon concentrations product (Chen et al., 2023; Xu et al., 2020). Its spatial resolution ($0.5° \times 0.625°$, approximately 50 km $\times$ 60 km) renders it more appropriate for large-scale investigations (such as Arctic or China), which is not suitable for the Tibetan Plateau. In our latest revision, we have removed the experimental section about black carbon and AOD, and have added the following content to the *Discussion 5.3*:

*Atmospheric pollutants, especially those referred to as light-absorbing aerosols, such as black carbon, brown carbon and dust, can warm the atmosphere (Kang et al., 2019; Ji et al., 2015). After being deposited onto snowpack, these light-absorbing particles can reduce the surface albedos of snowpack and promote its melting (Zhang et al., 2018; Lau et al., 2018). Despite the potential importance of these factors to the SP of TP, they were not analyzed in this study due to limited data availability of these factors over extended spatial and temporal scales. Developing high-resolution, spatiotemporal continuous datasets for these factors will be useful in future efforts to comprehensively quantify the response of SP to changing climate conditions.*

**References:**

Bai, K., Li, K., Ma, M., Li, K., Li, Z., Guo, J., Chang, N.-B., Tan, Z., and Han, D.: LGHAP: the Long-term Gap-free High-resolution Air Pollutant concentration dataset, derived via tensor-flow-based multimodal data fusion, Earth System Science Data, 14, 907-927, https://doi.org/10.5194/essd-14-907-2022, 2022.

Chen, X., Kang, S., Yang, J., and Hu, Y.: Contributions of biomass burning in 2019 and 2020 to Arctic black carbon and its transport pathways, Atmospheric Research, 296, https://doi.org/10.1016/j.atmosres.2023.107069, 2023.

Ji, Z., Kang, S., Cong, Z., Zhang, Q., Yao, T.: Simulation of carbonaceous aerosols over the third pole and adjacent regions: distribution, transportation, deposition, and climatic effects, Climate Dynamics, 45 (9), 2831–2846, https://doi.org/10.1007/s00382-015-2509-1, 2015.

Lau, W. and Kim, K.-M.: Impact of Snow Darkening by Deposition of Light-Absorbing Aerosols on Snow Cover in the Himalayas–Tibetan Plateau and Influence on the Asian Summer Monsoon: A

Possible Mechanism for the Blanford Hypothesis, Atmosphere, 9, https://doi.org/10.1007/10.3390/atmos9110438, 2018.

Xu, X., Yang, X., Zhu, B., Tang, Z., Wu, H., and Xie, L.: Characteristics of MERRA-2 black carbon variation in east China during 2000–2016, Atmospheric Environment, 222, https://doi.org/10.1016/j.atmosenv.2019.117140, 2020.

Xu, H., Ren, Y., Zhang, W., Meng, W., Yun, X., Yu, X., Li, J., Zhang, Y., Shen, G., Ma, J., Li, B., Cheng, H., Wang, X., Wan, Y., and Tao, S.: Updated global black carbon emissions from 1960 to 2017: improvements, trends, and drivers, Environment Science Technology, 55, 7869-7879, https://doi.org/10.1021/acs.est.1c03117, 2021.

Zhang, Y., Kang, S., Sprenger, M., Cong, Z., Gao, T., Li, C., Tao, S., Li, X., Zhong, X., Xu, M., Meng, W., Neupane, B., Qin, X., and Sillanpää, M.: Black carbon and mineral dust in snow cover on the Tibetan Plateau, The Cryosphere, 12, 413-431, https://doi.org/10.5194/tc-12-413-2018, 2018.

**Comment 2:**

The precipitation data is at high spatial resolution, which was deemed important and explaining differences with respect to earlier studies. Yet, the precipitation data consists of monthly means. Is this sufficient to elicit the daily snow evolution required to estimate snow onset and end dates? Is there daily precipitation dataset that the authors could have used and could they test if it affect the results?

**Response:** Given the large spatial extent of the Tibetan Plateau, there exists a significant time span encompassing both the snow onset date and snow end date, with snow onset date mainly occurring from September to December and snow end date ranging from February to August (Figure 1). This indicates that daily precipitation corresponding to snow onset/end date likewise crosses this identical temporal ranges. However, time series of precipitation often reveal natural periodicity and continuity, suggesting a time-dependent correlation in precipitation. Taking the example of snow onset date, starting from September 1st (as winter approaches), the precipitation exhibits a decreasing trend over time, indicating a negative temporal correlation (Hu et al., 2021). However, we found that the daily precipitation also exhibits a negative correlation ($R = -0.20$) with snow onset date using a daily meteorological dataset with 3 km resolution (Figure 2a). Ignoring the time dependence of precipitation data may lead to an overestimation of the negative impact of precipitation on snow onset date.

In fact, temperature is also time-dependent and provides a more intuitive understanding. Theoretically, colder temperatures are associated with earlier snow onset date, suggesting a positive correlation between temperature and snow onset date. However, our experiment based on the daily temperature corresponding to snow onset date shows a negative correlation ($R= -0.27$, Figure 2b). Temperature consistently decreases from September 1st onwards, displaying a negative correlation with time. Consequently, the actual effect of temperature on snow onset date may be obscured by its time dependence.

[Figure]

(a) Snow onset date             (b) Snow end date

**Figure1:** The spatial pattern of the multiyear averaged (a) Snow onset date, (b) snow end date on the TP from 2002 to 2022.

[Figure]

**Figure 2:** Scatter plot of snow onset date and precipitation (a), and temperature (b). DOS represents the day of the snow season, DOS 1 is equivalent to September 1.

Referred to literature on snow phenology (Chen et al., 2018; Ma et al., 2023), we defined the snow accumulation period (from September 1 to February 28/29) and the snow melting season (from March 1 to August 31), and calculate the average temperature and total precipitation for these two time periods, respectively. The average temperature and total precipitation from the snow accumulation season were used to analyze their impact on snow onset date. Similarly, the average temperature and total precipitation during the snow melting season were used to examine their influence on snow end date. The purpose of this is to address the time-dependence of meteorological factors by representing the general climatic conditions during the seasons of snow accumulation and melting. In the future, we intend to explore more methods to effectively isolate the time-dependency of meteorological factors.

We have added more discussion in *Section 5.1*:

*This study still has some limitations in exploring the effect of meteorological factors on SP. We calculated the average (or total) metrics of each meteorological factor during the snow accumulation season to explore their relationship with SOD using SEM. Similarly, we calculated the average (or total) metrics of each meteorological factor during the snowmelt season to investigate their relationship with SED (Chen et al., 2018; Ma et al., 2023). The purpose of this is to address the time dependence of meteorological factors. In the future, we intend to explore more methods for effectively separating the time dependence of meteorological factors when studying their effects on snow phenology.*

**References:**
Chen, X., Long, D., Liang, S., He, L., Zeng, C., Hao, X., and Hong, Y.: Developing a composite daily snow cover extent record over the Tibetan Plateau from 1981 to 2016 using multisource data, Remote Sensing of Environment, 215, 284-299, https://doi.org/10.1016/j.rse.2018.06.021, 2018.
Hu, W., Yao, J., He, Q., Chen, J.: Elevation-Dependent Trends in Precipitation Observed over and around the Tibetan Plateau from 1971 to 2017, Water, 13 (20), 2848. https://doi.org/10.3390/w13202848, 2021.
Ma, Q., Keyimu, M., Li, X., Wu, S., Zeng, F., and Lin, L.: Climate and elevation control snow depth and snow phenology on the Tibetan Plateau, Journal of Hydrology, 617, https://doi.org/10.1016/j.jhydrol.2022.128938, 2023.

**Comment 3:**

I am a bit unclear on the longitude dependence of the results is it merely elevation that is folded in this dependency (to explain that different longitudes receive different amount of radiation, precipitation and so forth).

**Response:** Indeed, the elevation of the TP decreases from west to east, resulting in a correlation between elevation and longitude. We conducted a correlation analysis between these two factors and found a correlation coefficient of 0.532 (Figure 3a). This correlation can cause the effect of longitude on SP folded under the effect of elevation on SP. However, controlling the influence of elevation, we still find the longitudinal dependency of snow cover. Taking the snow cover days as an example, at a fixed elevation (e.g., 5000 m), a correlation still exists between longitude and SCD ($R = 0.356$, Figure 3b). This implies that longitude could be interacting with additional factors to shape the spatial and temporal distribution of snow cover on the plateau.

[Figure]

**Figure 3:** Scatter plot of (a) elvation and longitude, (b) longitude and snow cover days (at a fixed elevation of 5000 m).

We have added more detail in *Discussion 5.3*:

*The elevation of the TP decreases from west to east, resulting in a correlation between elevation and longitude (R = 0.532, Figure 10a). This correlation can cause the effect of longitude on SP folded under the effect of elevation on SP. However, controlling the influence of elevation, we still find the longitudinal dependency of snow cover. Taking the SCD as an example, at a fixed elevation (e.g., 5000 m), a correlation still exists between longitude and SCD (R = 0.356, Figure 10b). This implies that longitude could be interacting with additional factors to shape the spatial and temporal distribution of snow cover on the plateau.*

**Minor comments:**

L 162: It is a bit unclear how the number 56 is derived. Number of stations times years with sufficiently long records? Please spell out the details.

**Response**: Since snow phenology is recorded annually, every station can calculate a valid yearly dataset for validation. However, not every station can extract snow phenology every year due to data gaps or other reasons. Therefore, the total number of records is not 24 stations multiplied by 19 years. Instead, the total number of records was extracted for all valid years for each station. We have added the following sentences in *Section 3.1*:

*Stations with fewer than 20 snow-covered days and fewer than 5 consecutive snow-covered days during the snow season were excluded. After applying these criteria, a total of 56 ground-observed SP from 24 stations were used for accuracy validation.*

L191: "owing to its valuable for unstable factor weights". Unclear sentence.

**Response:** For a better understanding, we have revised this sentence as follows:

*High multicollinearity between factors may result in inaccurate path coefficient, leading to these factors being misperceived as unimportant or invalid (Hair et al., 2010).*

L288: The wording "temperature gradient" is misleading (also caption of Fig 8). It is actually the correlation as a function of temperature that is shown, not the temperature vertical or spatial gradient (which would have different units).

**Response:** We have revised the word to "Temperature".

L326 Unclear sentence and wording for "essential": "proved consistently as essential  as elevation" (Is that what is meant here?)

**Response:** What we intend to convey here is that relative importance of temperature and precipitation shifts with elevation. We have also made revisions to this statement in *Section 5.1*:

*Overall, we identified that relative importance of temperature and precipitation shifts with elevation.*